# iControl3D: An Interactive System for Controllable 3D Scene Generation

Submission Id: 90

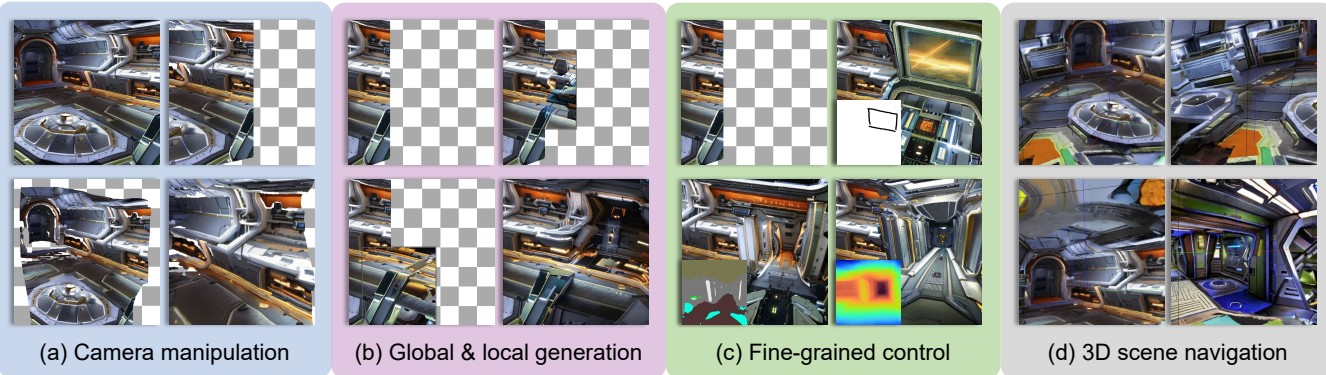

**Figure 1: Our system empowers users to generate and render customizable 3D scenes with precise control over the 3D scene generation process. With our system, users can actively participate in the 3D scene creation process. For example, they can (a) manipulate the virtual camera to any viewpoint, (b) adjust the size of the selection box to generate global and local content, and try different random seeds to generate various results. (c) Besides text prompts, users can achieve fine-grained control over the output by adding extra conditions such as scribbles, semantic segmentation maps, and depth. (d) After generating 3D scenes, they can navigate the entire scene and create camera trajectories to render videos according to their preferences.**

## ABSTRACT

3D content creation has long been a complex and time-consuming process, often requiring specialized skills and resources. While recent advancements have allowed for text-guided 3D object and scene generation, they still fall short of providing sufficient control over the generation process, leading to a gap between the user's creative vision and the generated results. In this paper, we present iControl3D, a novel interactive system that empowers users to generate and render customizable 3D scenes with precise control. To this end, a 3D creator interface has been developed to provide users with fine-grained control over the creation process. Technically, we leverage 3D meshes as an intermediary proxy to iteratively merge individual 2D diffusion-generated images into a cohesive and unified 3D scene representation. To ensure seamless integration of 3D meshes, we propose to perform boundary-aware depth alignment before fusing the newly generated mesh with the existing one in 3D space. Additionally, to effectively manage depth discrepancies between remote content and foreground, we propose to model remote content separately with an environment map instead of 3D meshes. Finally, our neural rendering interface enables users to build a radiance field of their scene online and navigate the entire scene. Extensive experiments have been conducted to demonstrate the effectiveness of our system.

## CCS CONCEPTS

• **Human-centered computing** → **Interactive systems and tools**; • **Computing methodologies** → *Computer vision*.

## KEYWORDS

Interactive user interface, 3D scene generation, controllable generation, mesh, neural rendering

## 1 INTRODUCTION

Recent years have witnessed explosive growth in the development of generative image and video models. In particular, diffusion models [14, 16, 39, 46] have pushed the boundaries of image generation, or AI-Generated Content (AIGC) to an unprecedented level of realism, with their outputs often indistinguishable from real images. Despite the success in the 2D domain, generating 3D assets and realistic 3D scenes remains a complex process that requires a significant amount of expertise and specialized software. It can take years of practice to master the necessary skills and techniques involved in the 3D content creation.

In light of this, many researchers are eager to extend the power of 2D diffusion models to the field of 3D generation. Existing works [18, 24, 26, 28–30, 33, 44, 50, 51, 56] have demonstrated the potential of text-guided 3D object generation using 2D diffusion. Yet, these methods present challenges when it comes to generating 3D structures and textures on a scene-scale level. Inspired by previous studies [5, 25, 27], Fridman et al. [15] introduce SceneScape, a novel method for text-driven perpetual view generation. While SceneScape enables the synthesis of flying-out trajectories of scenes from text, it struggles with generating complete 3D scenes. Concurrently, Text2Room [17] proposes to create room-scale textured 3D meshes by using pre-trained 2D text-to-image diffusion models. However, it is restricted to indoor scene generation and offers

limited control over the synthesis process, since only text and pre-defined camera trajectory are available. This can be frustrating for users who have specific creative visions for their 3D scene generation, as they cannot directly manipulate the scene's features or details to match their preferences.

In this paper, we present a novel system that can generate 3D scenes while providing users with fine-grained control over the creation process (see Fig. 1). Despite the existence of 3D generative models [4, 10, 54], the availability of large-scale 3D datasets required for their training is still limited. Motivated by prior works [15, 26], we instead rely on 2D diffusion models [39] that have been pre-trained on a large number of 2D images. For 3D scene generation, we use 3D meshes as an intermediary proxy to merge individual 2D images into a unified representation.

Our system builds upon a generative RGB-D fusion method. Specifically, we begin by obtaining an input image from the user or generating one using 2D diffusion. We then utilize a monocular depth estimator [3] to estimate the underlying geometry of the image and unproject it into 3D space to generate an initial mesh. After transforming the virtual camera to a new viewpoint, we render the mesh and apply 2D diffusion to inpaint holes and outpaint for new content. To ensure seamless integration of the generated content with the existing mesh, we estimate the depth of the image from that viewpoint and perform boundary-aware depth alignment. We then fuse the new mesh with the existing one in 3D space. The above process is repeated iteratively until we obtain a satisfactory complete 3D structure. However, outdoor scenes often pose challenges as 3D meshes cannot handle dramatic depth discontinuities well. To address this issue, we propose to model remote content (e.g., sky) separately with an environment map. This leads to more realistic outdoor scene representation.

To provide users with fine-grained control over the creation process, we develop a 3D creator interface that enables users to actively participate in the 3D scene creation process. Our interface offers several advantages. First, users can manipulate the virtual camera to any viewpoint and customize camera trajectories to create personalized 3D scenes. Second, users can adjust the size of the selection box to generate local content, and try different random seeds to generate various results. Third, inspired by ControlNet [53], we adopt a neural network structure to control diffusion models by adding extra conditions such as user scribbles, semantic segmentation maps, depth, and other information to achieve fine-grained control over the generation process. Finally, we introduce a neural rendering interface and incorporate Neural Radiance Fields (NeRFs) [31, 48] into our system, allowing users to create a radiance field of their scene online and navigate the entire scene. Users can also create camera trajectories to render videos according to their preferences.

In summary, our main contributions are:

- We present a new interactive system to generate and render customizable 3D scenes with user control. To this end, we introduce a 3D creator interface and a neural rendering interface.
- Our proposed boundary-aware depth alignment allows for the seamless integration of 3D meshes. To better handle outdoor scenes, we propose to model remote content with an environment map rather than 3D meshes.

- We achieve interactive 3D scene generation with precise controllability.

## 2 RELATED WORK

*3D-aware image synthesis.* Various 3D-GAN based methods [7, 8, 32, 42] have been proposed to combine neural scene representations with 2D generative models for 3D-aware image synthesis, enabling direct camera control. While these methods have demonstrated impressive results on the problem of generating single objects such as cars or faces, they are challenging to apply to large and diverse scenes. To extend 3D-aware image synthesis from single objects to completely unconstrained 3D scenes, several recent works [1, 13, 19, 45, 55] have been proposed. For example, GSN [13] proposes to break the radiance field into a grid of local radiance fields and collectively represent a scene by conditioning it on a 2D grid of floorplan latent codes. Bautista et al. [1] present GAUDI, where they first optimize a latent representation that disentangles radiance fields and camera poses, and then use the disentangled latent representation to learn a generative model. This allows for both unconditional and conditional generation of 3D scenes. However, these methods usually have a significant demand for extensive training and large-scale training data, limiting their generalization to only specific domains. Instead, our objective is to generate diverse 3D scenes.

*Perpetual view generation.* Perpetual view generation [20, 27] refers to the process of generating a continuous video sequence that corresponds to an arbitrary camera trajectory, using only a single image of the scene as input. Different kinds of methods have been explored in the literature. One line of research [23, 37, 49, 52] has focused on synthesizing indoor scenes with controllable camera trajectories. Motivated by Liu et al. [27], recent works such as InfNat-Zero [25] and DiffDreamer [5] aim at synthesizing fly-through videos of natural landscapes along long camera trajectories. Yet, due to their per-frame generation framework and the lack of underlying scene representations, these methods may suffer from issues such as domain drifting and inconsistent novel views. Recent studies [6, 12] learn a generative model for unconditional synthesis of unbounded 3D nature scenes with a persistent 3D scene representation. Although these methods are capable of producing view-consistent flythrough videos, they necessitate significant training on large-scale datasets and are restricted to a specific domain, e.g., landscapes. On the contrary, our system can generate diverse 3D scenes without the need for large-scale training.

*3D content generation.* Diffusion models [16, 35, 39, 40, 46, 47] have demonstrated remarkable success in generating highly realistic images and videos. By iteratively applying a series of steps, these models can transform a simple noise distribution into a complex, high-dimensional data distribution, resulting in images and videos that are virtually indistinguishable from real-world data. As diffusion models continue to advance and gain popularity in the 2D domain, researchers are exploring the possibility of using 2D diffusion priors to generate 3D content. Recent works [9, 18, 24, 26, 28, 29, 33, 44, 50, 51] have shown promise in

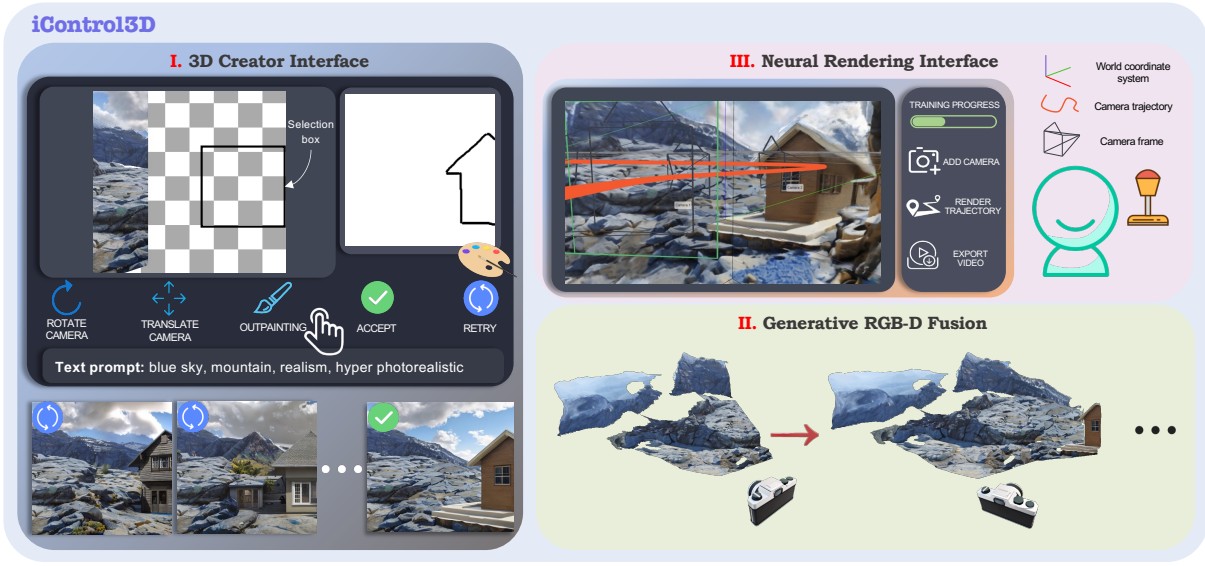

**Figure 2: System overview. (I) Within our 3D creator interface, users are allowed to manipulate the camera to any viewpoint, adjust the size of the selection box to generate local content, and try different random seeds to create a variety of results. Moreover, users can achieve fine-grained control over the generation process by adding extra conditions such as user scribbles; (II) Once the generated result in (I) is accepted by the users, our generative RGB-D fusion module fuses it with the existing mesh. This alternating process between (I) and (II) continues until a satisfactory 3D structure is obtained; (III) After generating 3D scenes, our neural rendering interface then builds a radiance field online and enables users to navigate the entire scene. By recording their virtual journey through the scene, users can also produce high-quality videos that showcase the intricacies and beauty of their designs.**

text-guided 3D object generation, but challenges remain in generating large-scale 3D structures and textures for entire scenes. Motivated by previous studies in perpetual view generation [5, 25, 27], Fridman et al. [15] propose SceneScape, a text-driven approach that synthesizes flying-out trajectories of scenes using 2D diffusion. However, SceneScape struggles with generating complete 3D scenes. Text2Light [11] introduces a zero-shot text-driven HDR panorama generation framework for creating 3D scenes but fails to impress users with freely moving cameras. Concurrently, Text2Room [17] uses pre-trained 2D text-to-image diffusion models to create textured 3D meshes of indoor scenes but offers limited control over the output. To bridge this gap, we present a novel system that can generate and render customizable 3D scenes with user control.

## 3 METHOD

### 3.1 System Overview

Our goal is to generate diverse 3D scenes while providing users with fine-grained control over the creation process. This entails tackling two challenges, i.e., leveraging 2D diffusion priors for consistent 3D scene generation and providing users with controllability over the creation process. Existing state-of-the-art methods [6, 12, 15, 17, 37] either require extensive training and fail to ensure global consistency or lack fine-grained control over the synthesis process. To achieve our goal, we present iControl3D, an interactive system for 3D scene generation with user control. We schematically illustrate our system in Fig. 2.

Our system mainly consists of a generative RGB-D fusion module, a 3D creator interface, and a neural rendering interface. Our system begins by obtaining an input image from users or generating one using 2D diffusion [39], estimating its geometry via a depth estimator [3], and generating an initial 3D mesh. We then render the mesh from different viewpoints, apply inpainting, perform boundary-aware depth alignment, and fuse it with the existing mesh, iteratively refining it until a satisfactory 3D structure is obtained. To handle outdoor scenes with depth discontinuities, we model remote content separately with an environment map, resulting in a more realistic representation. Unlike previous methods that offer a limited degree of user control, our system presents a 3D creator interface that enables users to actively participate in the 3D scene creation process. We also incorporate ControlNet [53], which can control diffusion models by adding extra conditions, into our interface to provide users with fine-grained control over the synthesized outputs. After generating 3D scenes, our neural rendering interface then builds a radiance field online and enables users to navigate the entire scene and create camera trajectories to render videos according to their preferences.

### 3.2 Generative RGB-D Fusion

**Initialization.** Motivated by previous works [15, 26], we leverage 2D diffusion models [39] that have been pre-trained on a large number of 2D images. Our system starts by obtaining an input image $I_0$ from users or generating one using 2D diffusion. Formally,

let $\mathcal{G}$ be a pre-trained 2D diffusion model. We then can generate the input image $I_0$ using 2D diffusion model $\mathcal{G}$:

$$I_0 = \mathcal{G}(T, z), \tag{1}$$

where $T$ is a text prompt and $z$ represents additional conditions, e.g., user scribbles, semantic segmentation maps, and depth maps. It is worth noting that 2D diffusion models can only generate independent 2D images without any 3D structural relationship between them. Hence, relying solely on 2D diffusion models is insufficient to create a unified 3D scene. Inspired by prior works [17], we leverage 3D meshes as an intermediary proxy to merge individual 2D images generated by 2D diffusion models into a unified 3D scene representation. To this end, we utilize an off-the-shelf monocular depth estimator [3] to estimate the underlying geometry of the input image. After that, we proceed to unproject the input image into an initial 3D mesh $\mathcal{M}_0 = (\mathcal{V}, \mathcal{F}, C)$ using depth values, where $\mathcal{V} = \{v_i\}_{i=1}^N$ is the set of $N$ vertices, $\mathcal{F} = \{f_i\}_{i=1}^F$ is the set of $F$ faces with each connecting three vertices, and $C = \{c_i\}_{i=1}^N$ are the color vectors attached on vertices.

**Mesh projection and inpainting.** We now have the initial 3D mesh $\mathcal{M}_0$. Our next step is to build up the scene iteratively. To do this, we generate new content from previously unobserved viewpoints. Specifically, we first render the mesh in the target camera pose $\mathbf{P}_{t+1}$:

$$\hat{I}_{t+1}, \hat{D}_{t+1}, \hat{m}_{t+1} = \Pi(\mathcal{M}_t, \mathbf{P}_{t+1}). \tag{2}$$

The mesh renderer $\Pi$ [36] produces the rendered image $\hat{I}_{t+1}$, the rendered depth $\hat{D}_{t+1}$ and the rendered mask $\hat{m}_{t+1}$ indicating the visible regions of the mesh in the rendered image, where pixels corresponding to visible and invisible parts of the mesh are set to 1 and 0, respectively. To create new content, the 2D diffusion model $\mathcal{G}$ is employed to inpaint missing pixels via

$$I_{t+1} = \mathcal{G}\left(\hat{I}_{t+1}, \sim\hat{m}_{t+1}, T, z\right), \tag{3}$$

where $\sim\hat{m}_{t+1}$ is the inverted mask used to guide the diffusion model by highlighting the areas of the image that should be inpainted.

**Boundary-aware depth alignment.** Likewise, we then employ the depth estimator to predict the underlying geometry of $I_{t+1}$, denoted as $\tilde{D}_{t+1}$. It should be noted that the depth of shared regions between the predicted depth map $\tilde{D}_{t+1}$ and the rendered depth map $\hat{D}_{t+1}$ may differ. To ensure seamless integration of the generated content with the existing mesh, it is intuitive to align the depth such that similar regions in a scene are placed at a similar depth as much as possible. This can help to avoid abrupt transitions at the boundaries between the generated content and the existing mesh. SceneScape [15] utilizes an online test-time training technique to promote the predicted depth map of the current frame to be in line with the geometric structure of the synthesized scene. However, this technique requires a certain amount of time to achieve depth alignment, making it unsuitable for real-time applications.

To this end, we propose boundary-aware depth alignment. The rationale behind incorporating boundary-aware depth alignment is to minimize the time needed for depth alignment, rendering it suitable for real-time 3D scene generation. This eliminates the necessity of prolonged waiting periods before progressing to the next step in scene creation. As shown in Fig. 3, we first obtain the

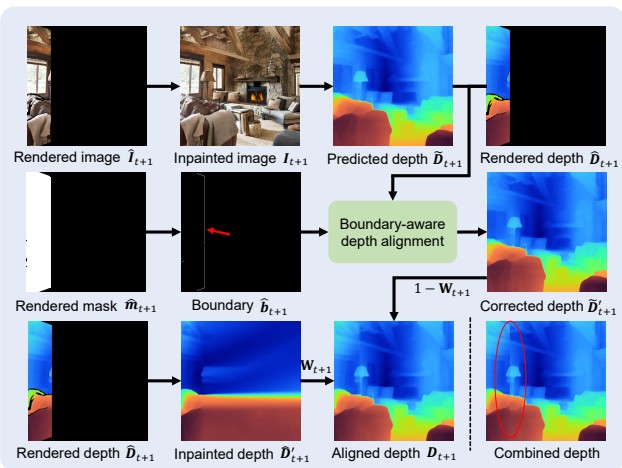

**Figure 3: Boundary-aware depth alignment. Directly combining the rendered depth $\hat{D}_{t+1}$ and the predicted depth $\tilde{D}_{t+1}$ leads to abrupt transitions in the combined depth while our boundary-aware depth alignment ensures a more seamless depth fusion.**

boundary $\hat{b}_{t+1}$ of the rendered mask $\hat{m}_{t+1}$ via:

$$\hat{b}_{t+1} = (\sim\hat{m}_{t+1} \oplus B) \cap \hat{m}_{t+1}, \tag{4}$$

where $\oplus$ and $\cap$ respectively denote the dilation and intersection operation, while $B$ represents a structuring element used for the dilation operation. Inspired by Liu et al. [27], we then conduct our boundary-aware depth alignment by solving the following least squares problem:

$$\min_{\alpha, \beta} \left\| \hat{b}_{t+1} \odot \left( \frac{\alpha}{\tilde{D}_{t+1}} + \beta - \frac{1}{\hat{D}_{t+1}} \right) \right\|^2, \tag{5}$$

where $\odot$ denotes the Hadamard (i.e., element-wise) product. Intuitively, this optimization attempts to scale and shift the predicted disparity such that the aligned disparity of the boundary region matches the rendered disparity. After obtaining the optimal scale and shift parameters, we can use them to compute the corrected depth $\tilde{D}'_{t+1}$ as follows:

$$\tilde{D}'_{t+1} = \frac{1}{\alpha/\tilde{D}_{t+1} + \beta}. \tag{6}$$

To ensure a smoother depth transition, we further propose a depth blending technique. Specifically, we first inpaint the rendered depth $\hat{D}_{t+1}$ with the Navier-Stokes inpainting algorithm [2], resulting in the inpainted depth $\hat{D}'_{t+1}$. Next, we blend the corrected depth $\tilde{D}'_{t+1}$ with the inpainted depth $\hat{D}'_{t+1}$ to compute the aligned depth $D_{t+1}$ as follows:

$$D_{t+1} = \mathbf{W}_{t+1} \cdot \hat{D}'_{t+1} + (1 - \mathbf{W}_{t+1}) \cdot \tilde{D}'_{t+1}, \tag{7}$$

where the weight map $\mathbf{W}_{t+1}$ is obtained by applying Gaussian blur to $\hat{m}_{t+1}$.

**Mesh fusion.** Given the aligned depth map $D_{t+1}$, our next step is to generate a new 3D mesh representation of the scene $\hat{\mathcal{M}}_{t+1}$ from the 2D image $I_{t+1}$ and fuse it with the existing mesh $\mathcal{M}_t$. We first

Table 1: Comparison of ours and relevant works. *Indoor scene*: Designed for handling indoor scenes. *Outdoor scene*: Designed for handling outdoor scenes. *No large-scale training*: Not requiring large-scale training. *Radiance field*: If radiance fields are used. *Interactive generation*: If interactive generation is supported using an interface. *Local generation*: If local generation is supported. *Conditional synthesis*: If the synthesis can be conditioned on additional input. *Text control*: If the generation can be controlled by text prompts. *Fine-grained control*: If having precise control over the generation process, e.g., scribbles.

| Method | Indoor scene | Outdoor scene | No large-scale training | Radiance field | Interactive generation | Local generation | Conditional synthesis | Text control | Fine-grained control |
|---|---|---|---|---|---|---|---|---|---|
| PixelSynth [38] | ✓ | ✗ | ✗ | ✗ | ✗ | ✗ | ✓ | ✗ | ✗ |
| InfNat-Zero [25] | ✗ | ✓ | ✗ | ✗ | ✗ | ✗ | ✓ | ✗ | ✗ |
| LOR [37] | ✓ | ✗ | ✗ | ✗ | ✗ | ✗ | ✓ | ✗ | ✗ |
| SceneScape [15] | ✓ | ✗ | ✓ | ✗ | ✗ | ✗ | ✓ | ✓ | ✗ |
| GSN [13] | ✓ | ✗ | ✗ | ✓ | ✗ | ✗ | ✓ | ✗ | ✗ |
| SGAM [43] | ✓ | ✓ | ✗ | ✗ | ✗ | ✗ | ✓ | ✗ | ✗ |
| SceneDreamer [12] | ✗ | ✓ | ✗ | ✓ | ✗ | ✗ | ✗ | ✗ | ✗ |
| Persistent Nature [6] | ✗ | ✓ | ✗ | ✓ | ✗ | ✗ | ✗ | ✗ | ✗ |
| Text2Room [17] | ✓ | ✗ | ✓ | ✗ | ✗ | ✗ | ✓ | ✓ | ✗ |
| NF-LDM [21] | ✓ | ✓ | ✗ | ✓ | ✗ | ✗ | ✓ | ✗ | ✗ |
| Ours | ✓ | ✓ | ✓ | ✓ | ✓ | ✓ | ✓ | ✓ | ✓ |

unproject the image pixels into 3D space using the camera intrinsic matrix and target camera pose $\mathbf{P}_{t+1}$. Once we have a set of 3D points representing the scene, we follow Höllein et al. [17] and use a triangulation scheme to construct a mesh representation $\hat{\mathcal{M}}_{t+1}$. This scheme involves connecting each set of four neighboring points in a regular grid pattern to form two triangles.

To fuse the new 3D mesh $\hat{\mathcal{M}}_{t+1}$ with the existing mesh $\mathcal{M}_t$, we extend the triangulation scheme at the edges of the inpainting mask $\sim \hat{m}_{t+1}$ to connect these faces with their neighboring faces from the existing mesh $\mathcal{M}_t$. This process results in the fusion of the two meshes, producing the final mesh $\mathcal{M}_{t+1}$.

**Environment map modeling.** We can repeat the aforementioned process iteratively until we achieve a satisfactory 3D structure. However, outdoor scenes may pose a challenge as 3D meshes struggle to handle dramatic depth discontinuities, e.g., between the sky and ground. These discontinuities often lead to flawed structures or large holes in the reconstructed mesh, leading to visible artifacts in the fusion of the new 3D mesh with the existing one. To this end, we propose to model remote content separately with an environment map. Specifically, we assume that remote content has an infinite depth and can be represented as a texture on a sphere surrounding the scene. To embed the remote region into the environment map, for each $I_{t+1}$, we use SAM [22] to segment the remote region in the image $I_{t+1}$ and map each pixel in the segmented region to a point on the surface of a sphere using inverse equirectangular projection. When we change to the next viewpoint, we first obtain the remote content from the environment map, followed by the mesh projection. This allows for accurate rendering of remote regions in subsequent steps, bypassing the issue of depth discontinuities between remote content and foreground.

### 3.3 3D Creator Interface

Current methods [6, 11, 12, 15, 17, 37] provide limited control over the synthesis process as they only allow for text and predefined camera trajectories as input. This can be frustrating for users who have specific creative visions or requirements for their 3D scene generation, as they cannot directly manipulate the scene's features or details to match their preferences. To overcome this limitation, we introduce a 3D creator interface as a key component of our system. The interface provides a user-friendly and intuitive way for users to actively participate in the 3D scene creation process. Our 3D creator interface offers several advantages (see Fig. 1). One of the most notable features of our interface is the ability for users to adjust the size of the selection box, allowing them to generate local content and try different random seeds to create a variety of results. This feature gives users the ability to select the best output that matches their creative vision. The virtual camera module is another highlight of our interface. It allows users to manipulate the camera to any viewpoint and customize camera trajectories, providing a personalized experience for creating 3D scenes.

**Fine-grained control.** Our objective is to provide users with not only full control over the 3D scene creation process but also fine-grained control to achieve their desired level of detail and customization. Inspired by ControlNet [53], we adopt a neural network structure to control diffusion models, which allows users to achieve fine-grained control over the generation process by adding extra conditions such as user scribbles, semantic segmentation maps, depth, and other information. This feature enables users to create more complex and detailed 3D scenes.

### 3.4 Neural Rendering Interface

Our generative RGB-D fusion module uses 3D meshes as an intermediary proxy to merge individual 2D images into a unified 3D scene representation. However, we do not employ hole-filling and smoothing techniques as used in previous works [17]. This is because we empirically find that iterative mesh reconstruction often leads to unavoidable artifacts in 3D meshes. We instead propose to leverage the 2D diffusion-generated images, which are usually visually pleasing. These images have shared a 3D structural relationship due to our generative RGB-D fusion module. We, therefore, introduce a neural rendering interface and integrate Neural Radiance Fields [31, 48] into our system to further smooth the artifacts shown in 3D meshes. We train a neural radiance field using the 2D diffusion-generated images and their corresponding poses. This enables users to create a radiance field of their scene online and navigate the entire scene during training and after training. Our

**Table 2: Quantitative comparisons. We show that our system outperforms all baselines in terms of both the Inception Score (IS) [41] and CLIP Score (CS) [34].**

| Method | IS ↑ | CS ↑ |
|---|---|---|
| LOR [37] | 1.77 | 20.96 |
| SceneDreamer [12] | 1.35 | 21.35 |
| Persistent Nature [6] | 1.33 | 28.20 |
| Text2Room [17] | 2.57 | 28.50 |
| Ours | **2.63** | **29.77** |

**Table 3: User study. We conduct a user study to compare our system against competitive methods. All methods are evaluated on the perceptual quality (PQ) of the imagery and scene diversity (SD). Here we only present pairwise comparison results between ours and baselines.**

| Comparison | PQ ↑ | SD ↑ |
|---|---|---|
| LOR [37] / Ours | 7.6% / **92.4%** | 1.7% / **98.3%** |
| SceneDreamer [12] / Ours | 29.5% / **70.5%** | 11.9% / **88.1%** |
| Persistent Nature [6] / Ours | 17.7% / **82.3%** | 13.8% / **86.2%** |
| Text2Room [17] / Ours | 18.6% / **81.4%** | 25.0% / **75.0%** |

neural rendering interface offers users an immersive way to explore their 3D creations and produce customized videos.

## 4 EXPERIMENTS

In this section, we present a comprehensive evaluation of our system on a diverse range of indoor and outdoor scenes and compare its performance with state-of-the-art methods both quantitatively and qualitatively. Additionally, we conduct a user study to better evaluate the effectiveness of our system. Finally, we perform an ablation study to justify our design choices.

### 4.1 Baselines

Table 1 presents a comparison of our method with other relevant works. In our experiments, we primarily compare ours against four representative works including LOR [37], SceneDreamer [12], Persistent Nature [6], and Text2Room [17]. Specifically, LOR [37] is an autoregressive method that can generate long-term 3D indoor scene video from a single image but presents challenges when it comes to generating consistent 3D structures and textures on a scene-scale level. SceneDreamer [12] and Persistent Nature [6] learn a generative model for unconditional synthesis of unbounded 3D nature scenes with a persistent 3D scene representation, but necessitate significant training on large-scale datasets and are restricted to a specific domain. Text2Room [17] uses pre-trained 2D text-to-image diffusion models to create textured 3D meshes of indoor scenes but lacks fine-grained control over the synthesis process.

Our method distinguishes Text2Room in four significant ways. The most notable difference from Text2Room is the introduction of an interactive system designed to facilitate the comprehensive creation of a 3D scene by the user. The key strength of this system lies in its capacity to empower users with finer control over generated content. It allows for the integration of text with other modalities such as scribbles and semantic segmentation maps, offering users the capability to select specific parts of the scene for focus. Secondly, while Text2Room employs scale-and-shift depth alignment, our method goes a step further by incorporating a depth blending technique around the boundary. This enhancement ensures a smoother depth transition in the generated scenes. Thirdly, Text2Room is limited to handling indoor scenes, whereas our method extends its capabilities to generate outdoor scenes through the incorporation of environment maps. This broadens the scope of scene generation possibilities beyond indoor environments. Furthermore, we integrate Neural Radiance Fields into our system to further smooth the artifacts shown in 3D meshes.

### 4.2 Results

**Evaluation metrics.** To evaluate our system, we utilize Inception Score [41] and CLIP Score [34] as our evaluation metrics. A higher Inception Score indicates that the generated images have both high quality and diversity, whereas a higher CLIP Score signifies a greater similarity between the generated image and the given text prompt.
**Quantitative comparisons.** We adopt 21 scene settings, including 6 challenging outdoor settings such as "mountain" and "garden", and 15 indoor settings such as "living room" and "spaceship", and randomly generate outdoor scenes twice and indoor scenes once, resulting in 12 outdoor scenes and 15 indoor scenes. Since our focus is not on achieving complete mesh reconstruction, we instead render 200 images to compute both Inception Score and CLIP Score for each scene. In our evaluation, we closely follow Text2Room. It employs 20 different trajectories for method evaluation, generating 60 images from novel viewpoints for each scene to calculate 2D metrics. Likewise, we adopt 21 scene settings, totaling 27 scenes for each method, and generate 200 images for each scene to compute both the IS and CS. Therefore, our evaluation scale aligns with that of Text2Room. As shown in Table 2, our system outperforms existing baselines, which indicates that our system produces high-quality and diverse images across different scene settings.
**Qualitative comparisons.** Fig. 7 and Fig. 8 present a qualitative comparison between our system and baselines. We showcase randomly extracted novel views of generated scenes. We find that LOR [37] exhibits the tendency to produce inconsistent novel views and susceptibility to error accumulation. These limitations can lead to domain drifting and a decline in output quality. While Scene-Dreamer [12] and Persistent Nature [6] can synthesize large camera trajectories consistently, they require extensive training and are limited to specific domains such as landscapes. On the other hand, Text2Room [17] performs well in indoor scenarios but faces challenges when dealing with outdoor scenes. It also often produces over-smoothed regions in the reconstructions. In contrast, our system can generate high-quality novel views in both indoor and outdoor scenes. In addition, we show in Fig. 4 that our system can achieve fine-grained control.

### 4.3 User Study

To further evaluate the performance of our system, we conduct a user study involving 65 participants with diverse backgrounds and expertise in the field. We use different approaches to generate 60 free-navigating videos of various scenes, respectively. To prevent

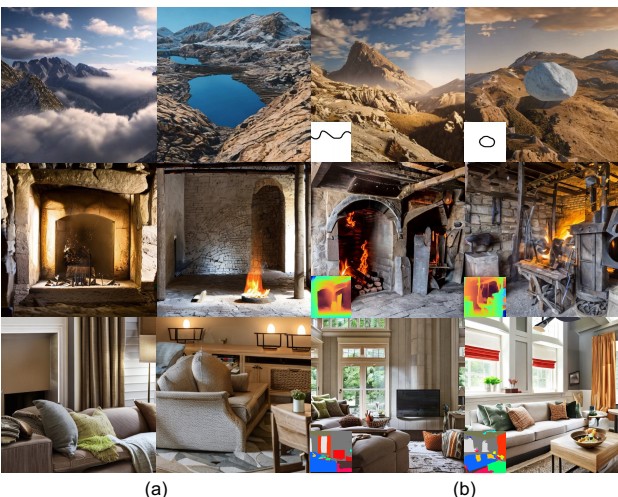

Figure 4: Fine-grained control. Compared to (a) current text-driven methods [15, 17], (b) our system can achieve fine-grained control over the output by adding extra conditions such as scribbles, depth, and semantic segmentation maps.

Table 4: Ablation study. We perform an ablation study on different components of our model to investigate their influence. Each component of our model contributes to the overall performance.

| Method | IS ↑ | CS ↑ |
| --- | --- | --- |
| w/o boundary-aware depth alignment | 2.19 | 28.37 |
| w/o environment map | 2.53 | 29.02 |
| Full model | **2.63** | **29.77** |

participants from guessing which results are generated by our system during the user study, we randomly present two sets of three videos each time. Both sets consist of three videos generated by randomly selected methods, rather than having one set generated exclusively by our system and the other set by another method. Participants are asked to compare two key aspects: the perceptual quality of the imagery and scene diversity. They are invited to choose the method with better perceptual quality and scene diversity, or none if difficult to judge. We report the results in Table 3, which points out that our system achieves higher perceptual quality and scene diversity compared to the alternative methods.

### 4.4 Ablation Study

To validate the effectiveness of each component of our system, we also conduct an ablation study. We design two variants of our system by removing boundary-aware depth alignment and environment map modeling while keeping the rest of the pipeline intact. As shown in Fig. 5 and Fig. 6, both boundary-aware depth alignment and environment map modeling contribute to the overall performance. Table 4 also confirms the effect of these components.

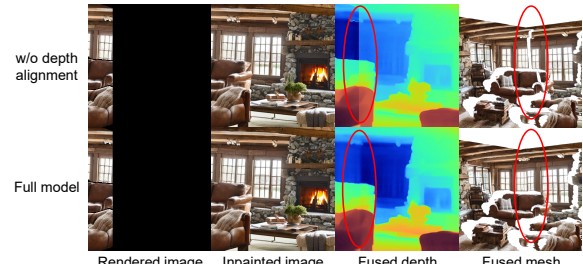

Figure 5: Effectiveness of boundary-aware depth alignment. Without boundary-aware depth alignment, the generated mesh may exhibit abrupt transitions at the boundaries between the newly generated content and the existing mesh.

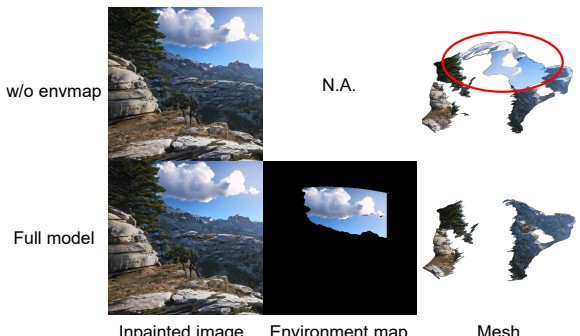

Figure 6: Effectiveness of environment map modeling. Without environment map modeling, handling outdoor scenes with 3D meshes becomes challenging due to dramatic depth discontinuities, leading to visible artifacts.

## 5 CONCLUSION

In this paper, we introduce iControl3D, an interactive system for controllable 3D scene generation and rendering. To achieve this, we develop a 3D creator interface to provide users with fine-grained control over the creation process and a neural rendering interface to allow them to navigate the entire scene. We show that our system can generate diverse 3D scenes with user control. We conduct extensive experiments to verify the effectiveness of our system. We hope that our system will inspire and empower users to unleash their creativity and bring their imaginations to life in the world of 3D content creation.

**Limitation.** While our method provides a user-friendly platform for interactive 3D content creation, certain challenges can impact its performance. One such challenge arises when the depth prediction module produces inaccurate geometry based on the input image, or when the segmentation model fails to predict with precision. These issues can compromise the quality of the generated 3D scenes. Moreover, distortions in the 3D meshes can further contribute to inaccuracies and inconsistencies, ultimately affecting the overall realism and quality.

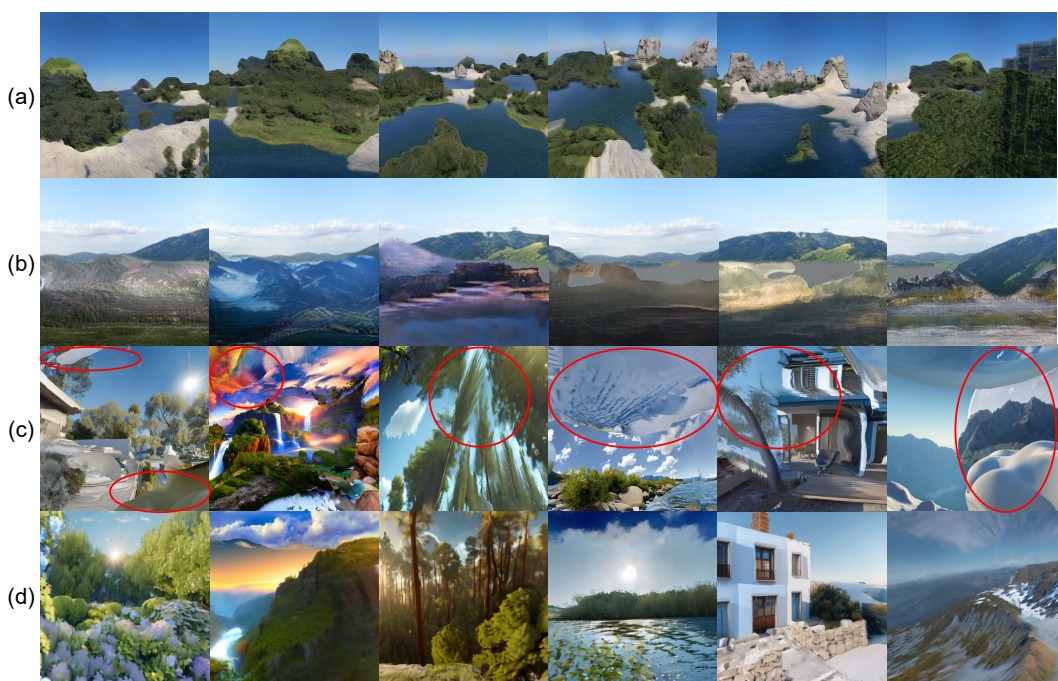

**Figure 7: Qualitative comparison on indoor scenes. Here we present the qualitative results of indoor scenes, displayed alternately from left to right. The scenes, in sequence, are "spaceship", "forge", "library", "cave", "ice castle", and "small office". (a) LOR [37], (b) Text2Room [17], and (c) ours. As can be seen, (a) LOR [37] is prone to domain drifting and a decline in output quality. Although (b) Text2Room [17] performs well on indoor scenes, it often produces over-smoothed artifacts in the reconstructions. In contrast, (c) our system presents diverse and photo-realistic results.**

**Figure 8: Qualitative comparison on outdoor scenes. We present the qualitative results of outdoor scenes, displayed alternately from left to right. The scenes, in sequence, are "garden", "waterfall", "forest", "river", "house", and "mountain". (a) Scene-Dreamer [12], (b) Persistent Nature [6], (c) Text2Room [17], and (d) ours. Note that (a) SceneDreamer [12] and (b) Persistent Nature [6] require extensive training and are limited to a specific domain, i.e., landscapes. While (c) Text2Room [17] can also generate outdoor scenes, it suffers from notable mesh distortions and artifacts. By contrast, (d) our system can generate high-quality and consistent novel views across diverse domains.**

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
