# OpenReview forum: "iControl3D: An Interactive System for Controllable 3D Scene Generation"
_acmmm.org/ACMMM/2024/Conference — MM2024 Poster_

### Official Review · Reviewer_h5UD · 2024-04-28

**Rating:** 5
**Confidence:** 3

**Summary:**

This paper introduces an interactive system to generate 3D scenes given 2D images. I gave credit for the seamless integration operation and the boundary-aware depth alignment operation. The experiments contain quantitative comparison, subjective comparison and ablation studies, which are sufficient personally. However, there are some questionable parts, exposition issues and lack of further discussions.

In general, my attitude is toward accepting this paper with minor revisions.

**Strengths:**

1. The proposed system is novel. I can see the engineering effort made by the authors.
2. The method to generate a sky box is interesting and practical.
3. The boundary-aware depth alignment operation is good.
4. The experiments are sufficient.
5. The paper is easy to read, and the supplementary video is good.

**Limitations:**

1. Line 35: I don't understand creative visions' meaning.
2. Line 105: I do not suggest adding so many references. We can pick a few typical ones.
3. Section 3.1, Lines 285-290: Such content is verbose personally. I believe the authors have discussed it in the introductions.
4. Equations 2 and 3: I am a bit confused. Does that mean an input image is converted to a mesh? The mesh is projected and used to generate a new image in the next iterations.
5. Line 402: What is the intuition of minimizing the time needed?
6. The boundary b_{t+1} is confusing. I suggest discussing its computational structure.
7. Does the dilation in Line 434 mean buffer operation in geometry?
8. I failed to find discussions of \alpha and \beta. I think we need to discuss Equation 5 more. I did not get the intuition of it.
9. I am fine with most of the experiments. However, I don't know the perceptual quality (Line 740). How do you define this term? How do you explain this term? How do users mark according to this term, e.g., rate 0-5? I think the authors should elaborate on the related content.
10. This paper introduced how we generate 3D scenes given 2D. Thus, I strongly recommend the authors discuss how their work differs from the literature that generates 3D given 3D. There are a few interactive scene synthesis frameworks similar to the proposed system, but they focus on manipulating 3D content (No 2D involved).

[1] The Clutterpalette: An Interactive Tool for Detailing Indoor Scenes LF Yu, SK Yeung, D Terzopoulos IEEE Transactions on Visualization and Computer Graphics.

[2] SceneDirector: Interactive Scene Synthesis by Simultaneously Editing Multiple Objects in Real-Time SK Zhang, H Tam, Y Li, KX Ren, H Fu, SH Zhang IEEE Transactions on Visualization and Computer Graphics.

[3] Mageadd: Real-time interaction simulation for scene synthesis SK Zhang, YX Li, Y He, YL Yang, SH Zhang Proceedings of the 29th ACM International Conference on Multimedia, 965-973.

**Suitability:**

3

---

### Official Review · Reviewer_umCL · 2024-05-20

**Rating:** 2
**Confidence:** 4

**Summary:**

The paper "iControl3D: An Interactive System for Controllable 3D Scene Generation" presents a novel system designed to empower users in generating and rendering customizable 3D scenes with precise control. The system utilizes 2D diffusion models to generate initial images, which are then transformed into 3D meshes through a monocular depth estimator. These meshes are iteratively refined by incorporating new viewpoints and inpainting missing regions. A boundary-aware depth alignment technique is introduced to ensure seamless integration of newly generated content with existing meshes. Additionally, the system includes a neural rendering interface that allows users to create and navigate radiance fields of their scenes.

**Strengths:**

- **Novelty**: The paper introduces a unique interactive system that combines 2D diffusion models with 3D mesh generation and neural rendering. This approach provides users with unprecedented control over the 3D scene creation process.
- **Technical Correctness**: The proposed methods, including the boundary-aware depth alignment and the use of 3D meshes as an intermediary, are well-founded and effectively address the challenges in 3D scene generation.
- **Adequate Evaluation**: The system's effectiveness is demonstrated through extensive experiments, including quantitative comparisons with state-of-the-art methods and a user study involving 65 participants. Metrics like Inception Score (IS) and CLIP Score (CS) are used to evaluate the quality and diversity of the generated scenes.
- **Clarity**: The paper is well-structured, with clear explanations of the methods and the system architecture. Figures and tables are used effectively to illustrate key concepts and compare results.
- **Applications**: The system has significant potential applications in fields like virtual reality, gaming, and content creation, where customizable and high-quality 3D scenes are crucial.

**Limitations:**

- **Dependence on Depth Estimation Accuracy**: The quality of the generated 3D scenes heavily depends on the accuracy of the depth prediction module. Inaccurate depth predictions can lead to distortions and inconsistencies in the final output.
- **Handling Outdoor Scenes**: While the system addresses some challenges in generating outdoor scenes through the use of environment maps, there can still be issues with dramatic depth discontinuities, which may lead to visible artifacts.
- **Computational Resources**: The iterative process of generating and refining 3D meshes can be computationally intensive, potentially limiting the system's real-time applicability and accessibility for users with limited hardware capabilities.

**Suitability:**

2

---

### Official Review · Reviewer_fE5p · 2024-05-23

**Rating:** 5
**Confidence:** 2

**Summary:**

This paper introduces iControl3D, an interactive system for generating and rendering customizable 3D scenes with precise user control. The system features a 3D creator interface, allowing users to engage in scene creation by manipulating the virtual camera, adjusting selection boxes, and experimenting with different random seeds to produce varied outcomes. iControl3D uses 3D meshes as intermediaries to merge individually generated 2D images into a cohesive 3D scene representation. To ensure seamless integration of 3D meshes, the paper proposes a boundary-aware depth alignment method. For handling depth discrepancies in outdoor scenes, remote content is modeled separately using an environment map instead of 3D meshes. The system also includes a neural rendering interface, enabling users to build a radiance field of their scene and navigate the entire space. Extensive experiments demonstrate the system's effectiveness, showcasing its ability to generate high-quality and diverse 3D scenes both indoors and outdoors.

**Strengths:**

1. iControl3D offers a 3D creator interface that allows users to actively participate in the creative process of 3D scene generation, enhancing the interactivity of creation.
2. iControl3D enables users to exert fine-grained control over the 3D scene generation process, including adjustments to the camera perspective, modifications to the selection box size, and experimentation with random seeds.
3. With boundary-aware depth alignment technology, the system achieves seamless integration of 3D meshes, enhancing the coherence of the scene.
4. iControl3D integrates neural rendering technology, allowing users to build and navigate an online radiance field of the scene, providing an immersive way to explore 3D scenes.
5. The system is not only suitable for indoor scenes but also capable of handling outdoor scenes, expanding the application range of 3D scene generation.

**Limitations:**

1. If the depth prediction module generates inaccurate geometric information based on the input image, it may affect the quality of the generated 3D scenes.
2. It would be better to discuss the time cost at each stage of iControl3D.
3. The evaluation metrics used in the paper, IS and CLIP score, do not account for the multi-view consistency of generated content.

**Suitability:**

3

---

### Official Review · Reviewer_7C4g · 2024-05-24

**Rating:** 4
**Confidence:** 2

**Summary:**

The paper presents iControl3D, an innovative interactive system designed for customizable 3D scene generation. The system addresses the challenge of creating 3D content, which traditionally requires specialized skills and extensive time. iControl3D leverages 2D diffusion models and 3D meshes to enable users to generate and render 3D scenes with fine-grained control. The system integrates a 3D creator interface and a neural rendering interface, allowing users to manipulate virtual cameras, adjust scene elements, and add extra conditions like scribbles and depth maps. The system's technical advancements include boundary-aware depth alignment and environment map modeling, enhancing the realism and coherence of generated scenes. Extensive experiments and user studies demonstrate the effectiveness and user-friendliness of iControl3D compared to existing methods.

**Strengths:**

+ User Control and Interactivity:  The system provides a high degree of control over the 3D scene creation process, allowing users to manipulate virtual cameras, adjust selection boxes, and add conditions such as scribbles and depth maps. This level of interactivity is a significant advancement over existing methods that offer limited control.
+ Neural Rendering Interface: The integration of Neural Radiance Fields (NeRFs) into the system enhances the visual quality of the generated scenes, enabling users to navigate and render high-quality videos of their 3D creations.
+ Extensive Evaluation: The paper provides comprehensive quantitative and qualitative comparisons with state-of-the-art methods. The results show that iControl3D outperforms existing approaches in terms of perceptual quality and scene diversity.

**Limitations:**

- Dependency on Depth Prediction Accuracy: The quality of the generated 3D scenes is heavily reliant on the accuracy of the depth prediction module. Inaccurate depth predictions can lead to compromised scene quality.
- Handling of Mesh Distortions: Despite the advancements, the system may still encounter distortions in the 3D meshes, affecting the overall realism and quality of the generated scenes.
- Scalability and Performance: The iterative nature of mesh generation and depth alignment might pose scalability challenges for generating very large or highly detailed scenes. The performance implications of these processes on real-time applications are not extensively discussed.
- Limited Academic Contribution: The paper appears to be more of a systematic integration effort rather than presenting novel academic contributions. While the integration of existing techniques into a cohesive system is valuable, the academic novelty might be perceived as limited.

**Suitability:**

3

---

### Meta-Review · Area_Chair_ag3s · 2024-07-01

**Recommendation:** Accept (Poster)
**Confidence:** 4

**Metareview:**

This paper was reviewed by three experts in the field. The recommendations are Borderline Accept, Weak Accept, Weak Accept, Weak Accept. The authors have addressed most of the concerns from reviewers. Reviewers still remain concerned about the technical contributions of this work, but given the quality of results the proposed algorithm generated and system-level contribution, the proposed method still has enough contribution. Therefore, the decision is to recommend the paper for acceptance to ACM Multimedia 2024.

Given the concerns raised by the reviewers, we recommend the authors to carefully read all reviewers' final feedback and revise the manuscript as needed. We congratulate the authors on the acceptance of their paper!